# Impact of Sarcopenia on Survival in Patients Treated with FOLFIRINOX in a First-Line Setting for Metastatic Pancreatic Carcinoma

**DOI:** 10.3390/jcm12062211

**Published:** 2023-03-13

**Authors:** Lisa Lellouche, Maxime Barat, Anna Pellat, Juliette Leroux, Felix Corre, Rachel Hallit, Antoine Assaf, Catherine Brezault, Marion Dhooge, Philippe Soyer, Romain Coriat

**Affiliations:** 1Gastroenterology and Digestive Oncology Unit, Cochin Hospital AP-HP, 75014 Paris, France; 2Department of Radiology, Cochin Hospital AP-HP, 75014 Paris, France; 3UFR de Médecine, Université Paris Cité, 75006 Paris, France; 4INSERM U1016, CNRS UMR8104, Institut Cochin, Université Paris Cité, 75006 Paris, France

**Keywords:** metastatic pancreatic carcinoma, FOLFIRINOX, sarcopenia, oxaliplatin

## Abstract

Sarcopenia, defined as decreased muscle mass and strength, can be evaluated by a computed tomography (CT) examination and might be associated with reduced survival in patients with carcinoma. The prognosis of patients with metastatic pancreatic carcinoma is poor. The FOLFIRINOX (a combination of 5-fluorouracil, irinotecan, and oxaliplatin) chemotherapy regimen is a validated first-line treatment option. We investigated the impact of sarcopenia on overall survival (OS) and progression-free survival (PFS) in patients with metastatic pancreatic carcinoma. Clinical data and CT examinations of patients treated with FOLFIRINOX were retrospectively reviewed. Sarcopenia was estimated using baseline CT examinations. Seventy-five patients were included. Forty-three (57.3%) were classified as sarcopenic. The median OS of non-sarcopenic and sarcopenic patients were 15.6 and 14.1 months, respectively (*p* = 0.36). The median PFS was 10.3 in non-sarcopenic patients and 9.3 in sarcopenic patients (*p* = 0.83). No differences in toxicity of FOLFIRINOX were observed. There was a trend towards a higher probability of short-term death (within 4 months of diagnosis) in sarcopenic patients. In this study, the detection of sarcopenia failed to predict a longer OS or PFS in selected patients deemed eligible by a physician for triplet chemotherapy and receiving the FOLFIRINOX regimen in a first-line setting, confirming the major importance of a comprehensive patient assessment by physicians in selecting the best treatment option.

## 1. Introduction

Pancreatic cancer is expected to become the second leading cause of cancer-related death by 2030 [1]. While surgical resection is the only potentially curative treatment, only 15–20% of patients are candidates for surgery at diagnosis, because the majority of patients are diagnosed at a locally advanced stage of the metastatic stage of the disease [2]. 

Gemcitabine was first identified as the cornerstone of the treatment of patients with metastatic pancreatic carcinoma [3]. In 2011 and 2013, two large phase 3 trials pinpointed a survival benefit with FOLFIRINOX (5-fluorouracil, irinotecan, and oxaliplatin) and gemcitabine plus nab-paclitaxel in comparison to gemcitabine monotherapy [4,5]. These combinations are now considered as the two validated options in the first-line setting for patients with metastatic pancreatic cancer, pending a good performance status (PS) (i.e., Eastern Co-operative Oncology Group [ECOG] PS 0 or 1). Despite these treatment improvements, the prognosis of patients with metastatic pancreatic adenocarcinoma is still poor [6]. 

Sarcopenia, defined as the decrease in skeletal muscle mass and strength, is a component of cancer cachexia, which is characterized by a negative protein and energy balance, resulting from multiple factors, such as reduced food intake, inflammation, and excessive catabolism [7,8]. In clinical practice, the most commonly used method for skeletal muscle mass assessment is obtained using cross-sectional imaging at the level of the third lumbar vertebra (L3), using computed tomography (CT) [9,10]. Skeletal muscle index (SMI) cut-offs based on gender and body mass index (BMI) to classify sarcopenia have been published [11,12]. Sarcopenia was significantly associated with a shortened overall survival (OS) (*p* < 0.001) and a reduced cancer-specific survival (CSS) (*p* < 0.001) in a large meta-analysis including 7843 patients with solid tumors [13]. At the time of diagnosis, the prevalence of sarcopenia in patients with solid tumors was estimated to be around 40% [12]. In pancreatic adenocarcinoma, the prevalence of sarcopenia ranges from 19 to 65% [12,14,15]. Recently, a Japanese study identified a shortened OS in sarcopenic patients treated with FOLFIRINOX for advanced pancreatic carcinoma (*p* = 0.001) [16].

The aim of this study was to determine whether sarcopenia was associated with an unfavorable outcome in a Western population of patients with metastatic pancreatic cancer treated with FOLFIRINOX in a first-line setting.

## 2. Material and Methods 

### 2.1. Study Design and Objectives 

We performed a single-center, retrospective study in patients with metastatic pancreatic carcinoma treated with a modified FOLFIRINOX regimen in the first-line treatment, from January 2012 to December 2020 in our tertiary center. The primary endpoint of the study was OS, defined as the time from diagnosis to death (or last news if alive). Secondary endpoint was PFS, defined as the time from diagnosis to radiological progression. Our study received approval from our local institutional review board (AAA-2022-08011). 

### 2.2. Patients and Treatment 

Patients were included in the study if they had a histologically proven diagnosis of metastatic pancreatic carcinoma and had received at least one cycle of a triplet chemotherapy with 5-fluorouracil, oxaliplatin, and irinotecan (FOLFIRINOX regimen). All patients received prophylactic growth factors to prevent severe neutropenia.

Patients were excluded if they did not have CT examination within the 30 days before the treatment initiation, if they did not have follow-up with CT examination, or if they had undergone a surgical resection or local treatment of the primary tumor or metastasis after the diagnosis of metastatic pancreatic cancer. Patients presenting with mixed tumors, or neuroendocrine tumors were excluded. Patients with metachronous metastasis were included in the present study.

### 2.3. Toxicity Assessment 

Treatment toxicity was evaluated during medical visit by experienced physicians after four to six cycles of chemotherapy and at progression. All side-effects were graded according to the Common Terminology Criteria for Adverse Events version (CTCAE) version 4 [17].

### 2.4. Anthropometric Measurement

For each patient, weight and height were measured according to standard methods, and body mass index (BMI) was calculated.

### 2.5. Image Analysis 

Sarcopenia was assessed using CT examination at the time of diagnosis of metastatic pancreatic cancer. A radiologist with 10 years of experience in pancreatic imaging analyzed CT images at the third lumbar vertebra (L3) and identified skeletal muscles according to anatomic features and predefined thresholds of Hounsfield units (−29 to +150) (Figure 1) [11]. Skeletal muscle area (cm^2^) was normalized by height (m^2^), allowing calculation of the skeletal muscle index (SMI) (cm^2^/m^2^). 

To define sarcopenia, we used the threshold values previously determined by Martin et al. which were associated with poor survival in patients with solid tumors [11]. Patients were considered sarcopenic when the following values were observed: SMI < 43 cm^2^/m^2^ for men with BMI < 25 kg/m^2^, <53 cm^2^/m^2^ for men with BMI ≥ 25 kg/m^2^, and <41 cm^2^/m^2^ for women, regardless of BMI. Radiologic progression was defined using the Response Evaluation Criteria In Solid Tumors (RECIST 1.1) criteria [18]. 

### 2.6. Statistical Analysis

The normality of the distribution of quantitative variables was assessed using Shapiro–Wilk test. Quantitative variables were expressed as means ± standard deviations (SD) and ranges when normally distributed, or as medians and interquartile ranges (Q1 and Q3) when non-normally distributed [19]. Qualitative variables were expressed as raw numbers, proportions, and percentages. Comparison between patients with sarcopenia and patients without sarcopenia was performed using Student t-test for continuous variables or the Chi2 test for qualitative variables. Survival in patients with sarcopenia and in patients without sarcopenia was analyzed by the Kaplan–Meier method and compared using the log-rank test. A *p*-value < 0.05 was considered to indicate significant differences. Calculations were performed with NCSSC 2007 software (NCSS, Kaysville, UT, USA).

## 3. Results 

### 3.1. Patients 

One hundred and seventy patients with histologically proven metastatic pancreatic carcinoma were initially identified. Among them, 24 were excluded due to the lack of a CT examination at the time of diagnosis, surgical resection of the primary tumor or metastasis (n = 3), or exclusive supportive care (n = 15). One hundred and twenty-eight patients (75.3%) received chemotherapy. Among them, 75 received a FOLFIRINOX regimen (58.7%), 33 received FOLFOX (25.8%), nine received gemcitabine plus nab-paclitaxel (7%), eight received gemcitabine monotherapy (6.2%), and three received FOLFIRI (2.3%). The study flow-chart is displayed in Figure 2. 

We included 75 patients who received at least one cycle of FOLFIRINOX. There were 38 women (50.7%) and 37 men (49.3%), with a mean age of 64 ± 11.2 (SD) years (range: 34–85 years). The patients’ baseline characteristics are reported in Table 1. All patients had a pancreatic ductal adenocarcinoma (PDAC) or variants (acinar cell carcinoma, n = 2; adenosquamous carcinoma, n = 1; undifferentiated carcinoma with osteoclast-like giant cells, n = 2). Ten patients had a past history of cephalic duodenopancreatectomy (n = 4) or pancreatosplenectomy (n = 6). Forty-three patients (57.3%) were identified as sarcopenic.

### 3.2. Toxicity 

In the overall population, 22.7% of patients (n = 17) experienced grade 3/4 hematological adverse events (Table 2). The most common grade 3/4 hematological adverse event was neutropenia or febrile neutropenia (12%) despite prophylactic treatment. Non-hematological grade 3/4 adverse events occurred in 26.7% of patients (n = 20), including diarrhea (n = 10), nausea (n = 5), vomiting (n = 3), and chemotherapy-induced neuropathy (n = 2). There were no significant differences regarding the adverse effects between sarcopenic and non-sarcopenic patients, except for anemia, which was significantly higher in non-sarcopenic patients. Oxaliplatin and irinotecan were discontinued in 44% and 18.7% of patients, respectively (Table 3). No significant differences were found in terms of treatment reduction or discontinuation between the two groups. 

### 3.3. Survival 

The median number of cycles of FOLFIRINOX administrated was 10 (range: 1–58) in the entire cohort, with no difference between sarcopenic and non-sarcopenic patients (9 vs. 10, *p* = 0.83). There were no significant differences in terms of the median OS (15.6 versus 14.1 months; 95% CI, 0.56–1.45; *p* = 0.36) or median PFS (10.3 vs. 9.3 months; 95% CI, 0.65–1.89; *p* = 0.83) between the non- and the sarcopenic patients (Figure 3). 

There were numerically more patients in the sarcopenic group who had an early death (25.6 versus 9.4%), within 4 months of diagnosis of metastatic pancreatic carcinoma, although this did not reach a statistical significance (*p* = 0.07) (Table 4). Seventy-two percent of sarcopenic patients who had a short-term death did not have a radiologically proven disease progression. There was no difference between the two groups in the percentage of deaths within 12 months of diagnosis.

At progression, 41.3% (n = 31) of patients received second-line chemotherapy (Table 5), and 58.7% (n = 44) received best supportive care. Sarcopenic patients received significantly less second-line chemotherapy than non-sarcopenic patients (30.2% vs. 56.3%, *p* = 0.02). The second-line treatment was gemcitabine monotherapy for 11 patients (14.7%) and gemcitabine plus nab-paclitaxel for 20 patients (26.7%). The median second OS (since the start of the second-line chemotherapy) for sarcopenic and non-sarcopenic patients was 12.3 months (5.9–16.6) in the patients receiving gemcitabine plus nab-paclitaxel, and 4.6 months (1.8–9.7) in the patients receiving gemcitabine monotherapy. 

## 4. Discussion 

Our study evaluated the association between sarcopenia at baseline and survival in 75 patients with metastatic pancreatic cancer who received FOLFIRINOX as the first-line therapy. We found no significant association between sarcopenia at baseline and OS or PFS. These findings are inconsistent with previous reports from Kurita et al. [16], who showed that sarcopenia at the time of diagnosis was an independent poor prognosis factor in 82 patients with advanced pancreatic cancer. There might be several explanations for these conflicting results. First, nearly half of the patients included in the study by Kurita et al. had previously received systemic therapy for advanced pancreatic cancer, whereas our study included only chemotherapy-naïve patients. Therefore, patients included in our study might have had a better general condition. Secondly, as the study by Kurita et al. involved an Asian population, the cut-offs used for the diagnosis of sarcopenia (SMI < 45.3 cm^2^/m^2^ and 37.1 cm^2^/m^2^ for men and women, respectively) were different than ours (SMI < 43 cm^2^/m^2^ for males with BMI < 25 kg/m^2^, <53 cm^2^/m^2^ for males with BMI ≥ 25 kg/m^2^, and <41 cm^2^/m^2^ for women, regardless of BMI). One difficulty in studying the impact of sarcopenia in clinical practice is the lack of consensus regarding the SMI thresholds for diagnosis. We chose to use those reported by Martin et al. in a large cohort of 1473 patients with lung or gastrointestinal tumors [11], but only 9.9% of the included patients had pancreatic carcinoma, the vast majority of them having a colon or rectum cancer. We might hypothesize that, because patients with pancreatic carcinoma suffer from cachexia more often than those with colon or rectal cancer, the SMI thresholds for the diagnosis of sarcopenia should be different. In another study including only obese patients with a lung or gastrointestinal cancer, Prado et al. found different sex-specific SMI cut-offs associated with mortality (52.4 cm^2^/m^2^ for men and 38.5 cm^2^/m^2^ for women) [9]. The narrative review by Bozzetti et al. reported that the cut-offs for defining sarcopenia ranged from 36 to 55 cm^2^/m^2^ in men [12].

In this study, we chose to include only patients who were treated with FOLFIRINOX, which is a validated first-line standard for patients with an ECOG PS of 0 or 1. As FOLFIRINOX is considered an aggressive regimen, it is recommended only for patients in good general condition based on the oncologist’s clinical assessment. In our center, of the 170 patients diagnosed with metastatic pancreatic cancer, only 44% ultimately received FOLFIRINOX, with the remaining patients receiving 5-fluorouracil-based bichemotherapy (21.2%), gemcitabine plus nab-paclitaxel (5.3%), gemcitabine monotherapy (4.7%), or exclusive support care (8.9%). Of the 75 patients receiving FOLFIRINOX, 11 (14.7%) and two (2.7%) had a reported ECOG PS of 2 and 3, respectively. These results should be interpreted with caution, as the literature reports conflicting data regarding the reproducibility of the ECOG PS scale [20,21]. In these 75 patients deemed eligible to receive FOLFIRINOX based on the physician global assessment, sarcopenia was not a predictor of reduced PFS of OS. 

In our study, sarcopenic patients received significantly less second-line chemotherapy than non-sarcopenic patients, although there was no difference in the median OS between the two groups. In metastatic pancreatic carcinoma, there are no large prospective randomized studies of second-line chemotherapy after FOLFIRINOX failure, as most data are from retrospective studies. In a prospective cohort of 57 patients receiving gemcitabine plus nab-paclitaxel after FOLFIRINOX failure, Portal et al. [22] identified a median second OS (since the start of the second-line chemotherapy) of 8.8 months. In our study, the median second OS in sarcopenic and non-sarcopenic patients treated with gemcitabine plus-paclitaxel was 12.3 months.

While sarcopenia at baseline was not a prognosis factor for OS or PFS in our study, there was a trend toward a higher proportion of early deaths (within 4 months of diagnosis of metastatic pancreatic carcinoma) in sarcopenic patients. This may argue for the early detection of sarcopenia in patients undergoing chemotherapy, to improve the overall management of patients and attempt to reverse skeletal muscle loss and cachexia. Recently, the APACaP trial randomized 313 patients with advanced pancreatic cancer, to chemotherapy or chemotherapy plus adapted physical activity (APA) [23]. In this trial, APA was shown to be feasible in patients with pancreatic carcinoma, and was associated with an improvement in several quality-of-life dimensions. Moreover, there was a tendency for a longer OS and PFS in the patients randomized to the APA arm, although this result did not reach a statistical difference. In a retrospective study including Japanese patients, Uemura et al. found that the baseline sarcopenia in patients with advanced pancreatic adenocarcinoma who received FOLFIRINOX was not associated with OS either [24]. However, these researchers did report the negative impact of an early decrease in skeletal muscle mass on the OS, which may indicate that, more than sarcopenia at diagnosis, maintaining muscle mass throughout treatment is an important factor for improving survival. 

Interestingly, the incidence of grade ≥ 3 adverse events was not significantly greater in patients with sarcopenia in our study. Anemia occurred surprisingly more often in patients without sarcopenia, but this should be interpreted with caution as we were unable to identify patients who underwent a blood transfusion or treatment with erythropoietin-stimulating agents. Various studies have reported an association between sarcopenia and chemotherapy toxicity [25,26,27]. More specifically, sarcopenic obesity has been associated with increased chemotherapy toxicity [27,28,29,30]. The administration of cytotoxic agents is usually determined by the body surface area (BSA), calculated from weight and height. It has been hypothesized that patients with obesity and sarcopenia would have a large BSA despite a low lean body mass. Therefore, sarcopenic obese patients would receive a high dose of chemotherapy despite a reduced volume of distribution [9]. We did not evaluate the impact of sarcopenic obesity on FOLFIRINOX tolerability in our study as we included only one obese sarcopenic patient. 

Our study has several limitations. First, it is a single-center study, which could have led to patient and treatment strategy selection bias. Second, it is a retrospective study with missing data, especially regarding the toxicity assessment. Finally, as discussed above, one of the major limitations to sarcopenia studies is the lack of consensus on the SMI threshold. To date, specific cut-offs for sarcopenia in patients with pancreatic cancer have not been reported in large studies or meta-analyses. 

## 5. Conclusions 

Sarcopenia at the time of diagnosis does not affect OS, PFS, or chemotherapy toxicity in selected patients receiving FOLFIRINOX for metastatic pancreatic carcinoma. Isolated sarcopenia should not be an exclusion criterion for the triplet chemotherapy regimen in patients deemed eligible by a comprehensive physician assessment. However, our results show a trend toward early death in sarcopenic patients, which should advocate for the early reversion of skeletal muscle loss as part of the global management of patients with metastatic pancreatic carcinoma. 

## Figures and Tables

**Figure 1 jcm-12-02211-f001:**
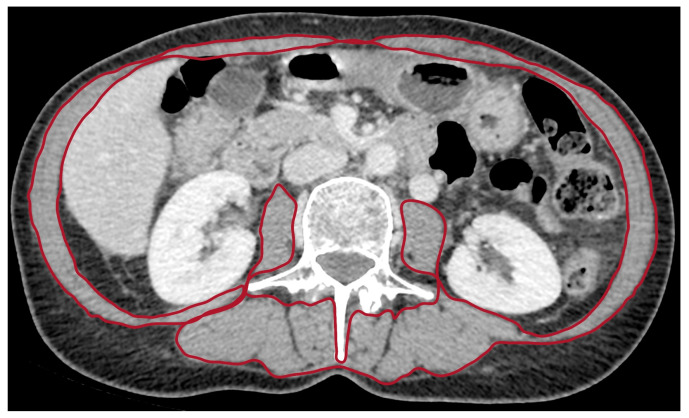
CT image in the axial plane at the level of the third lumbar vertebra in a sarcopenic patient with metastatic pancreatic carcinoma. Regions of interest (ROI) for sarcopenia measurements on axial CT image are indicated inside the red zone.

**Figure 2 jcm-12-02211-f002:**
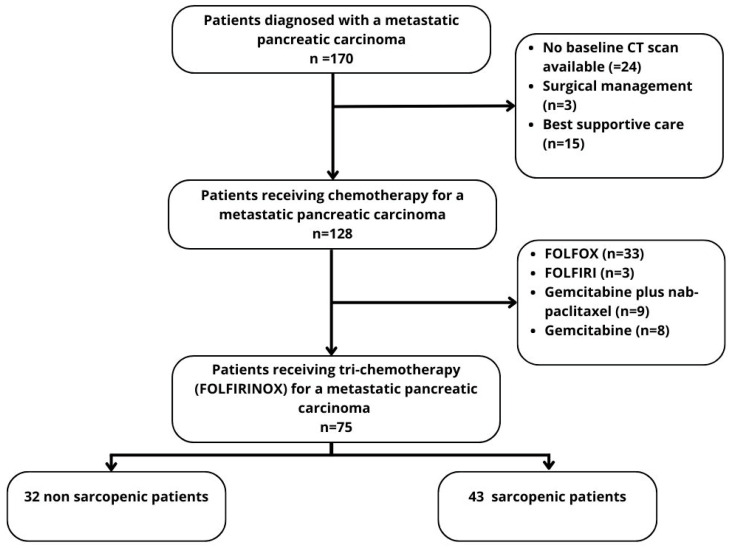
Flow-chart.

**Figure 3 jcm-12-02211-f003:**
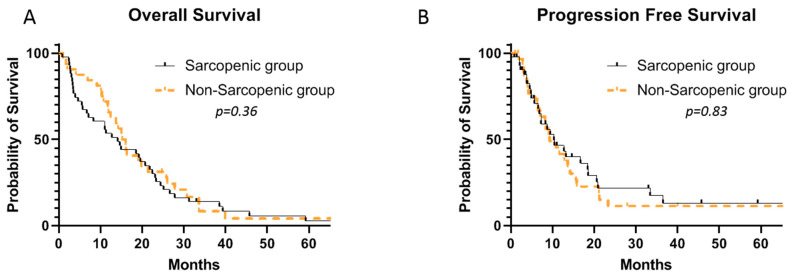
Median OS (**A**) and PFS (**B**) in sarcopenic and non-sarcopenic patients receiving FOLFIRINOX for metastatic pancreatic carcinoma.

**Table 1 jcm-12-02211-t001:** Patients’ characteristics at baseline.

Patients’ Characteristics	All Patients	Non-Sarcopenic Patients	Sarcopenic Patients	*p*-Value
Patients, n (%)	75	32 (42.7)	43 (57.3)	
Sex, n (%)				0.98
Women	38 (51)	13 (34.2)	25 (65)	
Men	37 (49.3)	19 (51.3)	18 (48.7)	
Age, mean (SD)	64 (11.2)	63.4 (11.7)	64.4 (11)	0.68
Weight (kg), mean (SD)	67.9 (13,5)	73 (12.4)	64 (13)	0.002
BMI, mean (SD)	23.6 (4.4)	25.3 (5.1)	22.3 (3.3)	0.005
Underweight (BMI < 18.5), n (%)	3 (4)	0 (0)	3 (7)	
Normalweight (18.5 ≤ BMI < 25), n (%)	49 (65.3)	18 (56.2)	31 (72)	
Overweight (25 ≤ BMI < 30), n (%)	17 (22.7)	10 (31.2)	7 (16.3)	
Obese (30 ≤ BMI), n (%)	4 (5.3)	3 (9.4)	1 (2.3)	
Skeletal muscle L3 area (cm^2^), mean (SD)	123.3 (31.5)	144.5 (24.5)	107.6 (26.7)	<0.001
SMI (cm^2^/m^2^) (men), mean (SD)	45.3 (7.5)	49.6 (4.7)	40.8 (7.3)	<0.001
SMI (cm^2^/m^2^) (women), mean (SD)	39.6 (9.1)	48.9 (8.8)	34.7 (4.3)	<0.001
ECOG PS, n (%)				0.2
0	14 (18.7)	8 (25)	6 (14)	
1	31 (41.3)	16 (50)	15 (34.9)	
2	11 (14.7)	3 (9.4)	8 (18.6)	
3	2 (2.6)	0 (0)	2 (4.6)	
Unknown	17 (22.6)	5 (15.6)	12 (28)	
Site of tumor, n (%)				0.36
Head	40 (53.3)	16 (50)	24 (55.8)	
Body or tail	33 (44)	16 (50)	17 (39.5)	
Unknown	2 (2.6)	0 (0)	2 (4.6)	
Biliary drainage, *n* (%)	19 (25.3)	9 (28)	10 (23.2)	0.6
Liver metastasis, n (%)	54 (72)	23 (71.9)	31 (72)	0.98
Pulmonary metastasis, n (%)	24 (32)	11 (34.4)	13 (30)	0.7
Peritoneum metastasis, n (%)	18 (24)	6 (18.7)	12 (28)	0.35
Previous therapy for localized cancer				
Adjuvant chemotherapy, n (%)	9 (12.0)	6 (18.7)	3 (7)	0.12
Pancreatic surgery, n (%)	10 (13.3)	6 (18.7)	4 (9.3)	0.23
CRP (mg/L), median (IQR)	13.6 (3.9–33.9)	13.5 (3.4–32.3)	15.1 (4.9–35.2)	0.8
Ca 19.9 (U/mL), median (IQR)	462.5 (48.9–3910.7)	425 (87.4–5179.5)	500 (39.2–1823)	0.08
CEA (ng/mL), median (IQR)	7 (3.8–29.7)	5.5 (3.3–12.6)	17.1 (6.3–36.6)	0.25
Total bilirubin (umol/L), median (IQR)	8.3 (5.5–16)	9.7 (6–16.5)	7.35 (5.3–15)	0.69
Albumin (g/L), median (IQR)	38.5 (34–41)	39 (36–42)	38 (33–40)	0.22

No differences were observed regarding tumor and metastasis localization, albumin, CRP, bilirubin, and CA 19–9 levels. Significant differences in mean BMI (22.3 kg/m^2^ vs. 25.3 kg/m^2^, respectively; *p* = 0.005) and mean weight (64 kg vs. 73 kg; *p* = 0.002) were found between sarcopenia and non-sarcopenia patients. Mean skeletal muscle L3 area (107.6 vs. 144.5; *p* < 0.0001) and SMI (34.7 vs. 48.9 for women and 40.8 vs. 49.6 for men; *p* < 0.0001) were significantly different between the two groups.

**Table 2 jcm-12-02211-t002:** Toxicity in patients receiving FOLFIRINOX regimen for metastatic pancreatic carcinoma.

	All Patients (n = 75)	Non-Sarcopenic Patients (n = 32)	Sarcopenic Patients (n = 43)	*p*-Value
**Neutropenia**				
Any grade	14 (18.6)	7 (21.9)	7 (16.3)	0.53
Grade ≥ 3	6 (8)	3 (9.4)	3 (7)	1
**Febrile neutropenia**				
Any grade	NA	NA	NA	
Grade ≥ 3	3 (4)	2 (6.2)	1 (2.3)	0.57
**Thrombopenia**				
Any grade	19 (25.3)	11 (34.4)	8 (18.6)	0.12
Grade ≥ 3	3 (4)	2 (6.2)	1 (2.3)	0.57
**Anemia**				
Any grade	30 (40)	17 (53)	13 (30)	**0.045**
Grade ≥ 3	5 (6)	2 (6.2)	3 (7)	1
**Diarrhea**				
Any grade	42 (56)	19 (59)	23 (53.5)	0.61
Grade ≥ 3	10 (13.3)	3 (9.4)	7 (16.3)	0.38
**Nausea**				
Any grade	34 (45.3)	17 (53)	17 (39.5)	0.24
Grade ≥ 3	5 (6.7)	2 (6.2)	3 (6.9)	0.90
**Vomiting**				
Any grade	21 (28)	9 (28.1)	12 (27.9)	0.98
Grade ≥ 3	3 (4)	1 (3.1)	2 (4.7)	1
**Peripheral neuropathy**				
Any grade	51 (68)	24 (75)	27 (62.8)	0.26
Grade ≥ 3	2 (2.6)	2 (6.2)	0 (0)	0.17

NA: Not applicable.

**Table 3 jcm-12-02211-t003:** Treatment interruption and dose reduction.

	All Patients n = 75	Non-Sarcopenic Patients n = 32	Sarcopenic Patients n = 43	*p*-Value
**Oxaliplatin, n (%)**				
Dose reduction	38 (50.7)	18 (56.2)	20 (46.5)	0.4
Discontinuation	33 (44)	15 (46.9)	18 (41.9)	0.66
**Irinotecan, n (%)**				
Dose reduction	29 (38.7)	10 (31.2)	19 (44.2)	0.25
Discontinuation	14 (18.7)	5 (15.6)	9 (20.9)	0.55
**5-fluorouracil, n (%)**				
Dose reduction	14 (18.7)	5 (15.6)	9 (20.9)	0.55
Discontinuation	0	0	0	NA

**Table 4 jcm-12-02211-t004:** Early death in patients receiving FOLFIRINOX for metastatic pancreatic carcinoma.

	All Patients (n = 75)	Non-Sarcopenic Patients (n = 32)	Sarcopenic Patients (n = 43)	*p*-Value
** *Death within 4 months from diagnosis, n (%)* **				
Death before confirmed radiological progression	10 (13.3)	2 (6.2)	8 (18.6)	0.08
Death before or after confirmed radiological progression	14 (18.6)	3 (9.4)	11 (25.6)	0.07
** *Death within 12 months from diagnosis, n (%)* **				
Death before confirmed radiological progression	12 (16)	4 (12.5)	8 (18.6)	0.54
Death before or after radiological progression	32 (42.7)	11 (34)	21 (48)	0.24

**Table 5 jcm-12-02211-t005:** Treatment at progression after FOLFIRINOX.

	All Patients n = 75	Non-Sarcopenic Patients n = 32	Sarcopenic Patients n = 43	*p*-Value
**Best supportive care**	44 (58.7)	12 (43.7)	30 (69.8)	**0.02**
**Second-line therapy**	31 (41.3)	18 (56.3)	13 (30.2)	**0.02**
Gemcitabine monotherapy, n (%)	11 (14.7)	6 (18.7)	5 (11.6)	0.38
Gemcitabine plus nab-paclitaxel, n (%)	20 (26.7)	12 (37.5)	8 (18.6)	0.06

## Data Availability

Data are available.

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
