# Peer review of "Impact of Sarcopenia on Survival in Patients Treated with FOLFIRINOX in a First-Line Setting for Metastatic Pancreatic Carcinoma"

_jcm, 2023, doi:10.3390/jcm12062211_

Round 1

Reviewer 1 Report

This is a very interesting article providing information of sarcopenia in pancreatic cancer.

There are some major issues which need to be addressed.

1.      There is no statistical difference of baseline ECOG PS between sarcopenia and non-sarcopenia. However, there are more ECOG 0/1 in non-sarcopenic group (75% vs 49%) and it is hard to believe that patients with ECOG 2 and 3 received FOLFIRINOX and tolerated it. This should be discussed in detail.

2.      Table 3 showed treatment interruption and dose reduction. It needs to clarify if starting dose of FOLFIRINOX (full dose or modified dose) is same for all patients.

3.      Table 5 showed significant number of non-sarcopenic patients received 2nd line therapy compared with sarcopenia (P=0.02) and there is no OS difference between two groups. This should be discussed in detail.      

Author Response

Dear reviewers,
We would like first to thank reviewers for their expertise on our manuscript and for their propositions of improvement.
We paid particular attention to all reviewers’ comments. Some references have been added in the revised manuscript.
The text was modified to answer all comments from the editor and the reviewers.
Please find below a point-by-point response to reviewer.
Yours sincerely

Reviewer 2 Report

Although the effect sizes were generally small, no power calculation prior to the inclusion is mentioned. It is possible that a type II error occurred. I understand that it will be hard to include more patients to address this, but an alternative approach could be to test a weakly prognostic variable in this cohort and validate that indeed power is a problem (or not). 

The above comment is especially pertinent in light of Table 4, where a very important difference is found between the groups that does not reach significance threshold. 

What is the outcome in sarcopenic patients treated with non-FFX as first line therapy in the center?

Table 5; why did the sarcopenia patients receive less 2nd line therapies? Did performance status deteriorate more in these patients?

Author Response

(The authors gave the same response as above.)

Reviewer 3 Report

This is a retrospective study aimed to determine whether sarcopenia is associated with an unfavorable outcome in a series of patients with metastatic pancreatic cancer.

There is current concern on this issue. However, it is surprising that the results of the study seem to contradict what literature suggests on the matter.

In general, the introduction is well contextualized, the statistics are correct and the results are well presented. 

Some suggestions to improve the quality of the manuscript:

Material & Methods:

-          Was muscular area measured with some specific software?

-          What differences in overall survival were expected between the two groups? Sample size should depend on this factor.

-          Please indicate the statistical software used for data analysis.

Results:

-          Please report the percentage of weight loss that patients had in the last 6 months.

-          The prognostic value of other nutritional variables (albumin, BMI, % weight loss…) could be analyzed.

-          Cutoff point for sarcopenia as proposed by Kurita Y et al could be analyzed, as it could have prognostic value in this series.

-          It would be interesting to explore the prognostic value of early body mass changes during treatment.

Discussion:

-          The authors try to explain the negative results of the study, but do not focus on what these results suggest: that sarcopenia is not a useful predictor of survival in this population. If the authors do not agree with their results, i.e., if they reject the null hypothesis, they should explain why.

-          Insufficient sample size should be mentioned as a possible explanation for the lack of differences in survival.

Author Response

(The authors gave the same response as above.)

Round 2

Reviewer 2 Report

I would have preferred to see my concerns addressed in a bit more depth. 

Author Response

Dear reviewer, 

Thanks a lot for your comment saying, "I would have preferred to see my concerns addressed in a bit more depth." 

We completely agreed with it and considered our study has a first step study. In fact, considering our study is a retrospective study some data were not sufficient to confirm the real impact of sarcopenia in pancreatic adenocarcinoma. 

We planned a second study to confirm our results. Second study will be a prospective study. 

Yours sincerely